# A Bayesian Nonparametrics View into Deep Representations

**Michał Jamroż**[*]
AGH University of Science and Technology
Krakow, Poland
mijamroz@agh.edu.pl

**Marcin Kurdziel**[*]
AGH University of Science and Technology
Krakow, Poland
kurdziel@agh.edu.pl

**Mateusz Opala**
AGH University of Science and Technology
Krakow, Poland
mo@matthewopala.com

## Abstract

We investigate neural network representations from a probabilistic perspective. Specifically, we leverage Bayesian nonparametrics to construct models of neural activations in Convolutional Neural Networks (CNNs) and latent representations in Variational Autoencoders (VAEs). This allows us to formulate a tractable complexity measure for distributions of neural activations and to explore global structure of latent spaces learned by VAEs. We use this machinery to uncover how memorization and two common forms of regularization, i.e. dropout and input augmentation, influence representational complexity in CNNs. We demonstrate that networks that can exploit patterns in data learn vastly less complex representations than networks forced to memorize. We also show marked differences between effects of input augmentation and dropout, with the latter strongly depending on network width. Next, we investigate latent representations learned by standard $\beta$-VAEs and Maximum Mean Discrepancy (MMD) $\beta$-VAEs. We show that aggregated posterior in standard VAEs quickly collapses to the diagonal prior when regularization strength increases. MMD-VAEs, on the other hand, learn more complex posterior distributions, even with strong regularization. While this gives a richer sample space, MMD-VAEs do not exhibit independence of latent dimensions. Finally, we leverage our probabilistic models as an effective sampling strategy for latent codes, improving quality of samples in VAEs with rich posteriors.

## 1 Introduction

Neural networks that differ only in initial parameter values converge to different minima of the cost function. This observation raises a following question: is this variability simply a manifestation of a numerical leeway afforded by model overparametrization or, perhaps, a manifestation of a more fundamental discord in ways neural networks take to make predictions? This question is not only important from a practical perspective – e.g. in efforts to pinpoint and interpret factors behind specific network responses – but is also fundamental to our understanding of information processing in neural models. Recently, Raghu et al. [2017] Morcos et al. [2018] and Kornblith et al. [2019] showed that under suitable similarity metric neural representations do in fact share some common structure. Yet, their work is limited to finding representational similarity between pairs of converged networks.

---

[*]Equal contribution.

In this article we aim to go beyond pairwise similarities and characterize neural representations from a probabilistic perspective. Specifically, we focus on two goals: characterizing sets of representations that are effectively reachable by convolutional networks and uncovering structure in latent spaces learned by variational autoencoders. To construct such characterizations we adopt Dirichlet Process Gaussian Mixture Models (DP-GMMs) as density models for deep representations. We then leverage tractable quantities in DP-GMMs to compare neural models with respect to the sets of representations they learn. Our main contributions are: (1) we propose probabilistic models for neural representations and use them to characterize sets of learned representations, (2) we show that memorizing nets learn vastly more complex representations than network trained on real data, (3) we demonstrate markedly different effects of two common forms of regularization on the complexity of learned representations and (4) we characterize latent spaces learned by $\beta$-VAEs and MMD-VAEs, demonstrating marked differences in representational capacity of their aggregated posteriors.

## 2 Dirichlet Process Mixture Model for neural representations

Our main idea in this work is to investigate neural representations using nonparametric mixture models. These flexible density models naturally adapt to the complexity of the underlying data distribution. We therefore leverage them as a principled way to quantify and compare complexity of representations learned by neural networks and to investigate latent representations in generative models. The specific nonparametric model we decided to use in this work, namely DP-GMM, was chosen because certain quantities of interest to us – e.g. when studying independence of dimensions in latent codes – are tractable in this model. Furthermore, it is consistent in total variation for distributions that are in the KL support of the prior and – assuming that the approximated density is sufficiently smooth – has near minimax contraction rate [Ghosal and Van der Vaart, 2017, sections 7.2 and 9.4].

We use DP-GMM to model representations learned by kernels in convolutional neural networks and to capture distributions of latent codes in variational autoencoders. In the latter case we take a learned inference distribution $q_\phi(\boldsymbol{z} \mid \boldsymbol{x})$ and construct a model for the *aggregated posterior*:

$$q_\phi(\boldsymbol{z}) = \int_{\boldsymbol{x}} q_\phi(\boldsymbol{z} \mid \boldsymbol{x}) p(\mathbf{x}) \, d\boldsymbol{x}. \tag{1}$$

Therefore, the set of observations in DP-GMM is simply the set of latent codes inferred for test images.

When modelling representations learned by CNN kernels, or *neurons' representations*, we use a construction similar to the one employed in Raghu et al. [2017], Morcos et al. [2018]. Consider a single convolutional kernel $k$ in a CNN layer. To construct a representation of the feature learned by $k$, we take a fixed sequence of input images $[\mathbf{x}_1, \mathbf{x}_2, \ldots, \mathbf{x}_l]$ and calculate a sequence of kernel responses: $[k(\mathbf{x}_1), k(\mathbf{x}_2), \ldots, k(\mathbf{x}_l)]$. These responses form a volume with shape $l \times h \times w$, where $h$ and $w$ are height and width of the layer output, respectively. We then perform an average pooling across spatial dimensions, obtaining an $l \times 1$ vector $\mathbf{a}_k$ that can be interpreted as a fingerprint of the feature learned by $k$, i.e. a neuron's representation:

$$\mathbf{a}_k = [\mathrm{avg\_pool}(k(\mathbf{x}_1)), \mathrm{avg\_pool}(k(\mathbf{x}_2)), \ldots, \mathrm{avg\_pool}(k(\mathbf{x}_l))]. \tag{2}$$

We repeat this procedure for every kernel in the given layer, using same input sequence in each case. However, unlike recent works on similarity of neural representations [Raghu et al., 2017, Morcos et al., 2018, Kornblith et al., 2019], we do not seek to find a transformation between two sets of neuron representations (i.e. between a pair of conv layers) that maximizes their similarity. We instead treat each learned representation $\mathbf{a}_k$ as a realization of a random variable that follows the distribution of representations learned in the given network layer. Under this interpretation we train multiple networks with identical architectures and hyper-parameter values, but different random initializations. Finally, we pool together representations learned by these networks[1]. Given $n$ trained networks and a convolutional layer with $m$ kernels, the set of DP-GMM observations therefore consist of $n \cdot m$ representations: $\{\mathbf{a}_1, \mathbf{a}_2, \ldots, \mathbf{a}_{n \cdot m}\}$.

There are two important aspects to our setup for convolutional networks. On the technical side, it is invariant with respect to the ordering of kernels in convolutional layers – any information about

initial ordering of kernels in a conv layer is lost in the set of observations modelled by DP-GMM. More importantly, this setup does not attempt to model a set of representations learned by a specific network instance. Rather, we want to capture the distribution of representations that are effectively reachable by a given layer in a certain network architecture and under certain training regime. This can be seen as capturing a restricted form of the notion of *effective capacity* formalized in Arpit et al. [2017]. That is, we can compare different networks and training regimes with respect to the sets of representations that are effectively learned under stochastic gradient descent.[2].

In the following sections we outline the DP-GMM formulation used in this work and explain how we employ it to quantify representational complexity in CNNs and investigate latent spaces in VAEs.

## 2.1 Generative model

Let $\mathcal{D} = \{\boldsymbol{x}_1, \boldsymbol{x}_2, \ldots, \boldsymbol{x}_N\}$ be a dataset of $N$ samples from some unknown $D$-dimensional probability distribution. To construct a density model for this distribution, we postulate a following generative model for $\boldsymbol{x}$:

$$\begin{aligned}
\alpha &\sim Gamma(1, 1), \\
G \mid \alpha &\sim DP(NIW(\boldsymbol{\theta}_0), \alpha), \\
\boldsymbol{\mu}_k, \boldsymbol{\Sigma}_k &\sim G, \\
\boldsymbol{x} \mid \boldsymbol{\mu}_k, \boldsymbol{\Sigma}_k &\sim \mathcal{N}(\boldsymbol{\mu}_k, \boldsymbol{\Sigma}_k).
\end{aligned} \tag{3}$$

Shortly, observations are assumed to come from a mixture of Gaussian components. Component parameters have a Dirichlet Process prior with concentration $\alpha$ (also uncertain, i.e. a model parameter with $Gamma(1, 1)$ prior). $G$ in this formulation stands for a random measure over components and their parameters. The base distribution in the Dirichlet Process, i.e. prior over the component mean $\boldsymbol{\mu}_k$ and covariance $\boldsymbol{\Sigma}_k$, is chosen to be a Normal-inverse-Wishart (NIW) distribution with hyper-parameters $\boldsymbol{\theta}_0$:

$$p(\boldsymbol{\mu}_k, \boldsymbol{\Sigma}_k) = NIW(\boldsymbol{\mu}_k, \boldsymbol{\Sigma}_k \mid \boldsymbol{\theta}_0), \quad \boldsymbol{\theta}_0 = \{\mathbf{m}_0, \nu_0, \kappa_0, \boldsymbol{S}_0\}. \tag{4}$$

We explain the choice of these hyper-parameters in Appendix A.

We use the Chinese Restaurant Process (CRP) as a constructive definition of the Dirichlet Process prior. Shortly, CRP describes a process of either assigning an observation to an existing component or creating a new component for it. In particular, let $c_i \in \boldsymbol{c} = \{c_1, c_2, \ldots, c_N\}$ be a component for the observation $\boldsymbol{x}_i$ and assume that vector of component assignments for other observations, denoted by $\boldsymbol{c}_{-i} = \boldsymbol{c} \setminus \{c_i\}$, is known. Then the probability $c_i = k \mid \boldsymbol{c}_{-i}$ under CRP is given by:

$$p(c_i = k \mid \boldsymbol{c}_{-i}, \alpha) = \begin{cases} \frac{N_{k,-i}}{\alpha + N - 1} & \text{if component } k \text{ exists,} \\ \frac{\alpha}{\alpha + N - 1} & \text{if } k \text{ is a new component,} \end{cases}$$

where $N_{k,-i}$ is the number of observations already assigned to the $k$-th component. This mechanism effectively puts a prior on the number of mixture components, making it a model parameter. The choice of NIW prior over component parameters is also significant. NIW is a conjugate prior to the multivariate Normal likelihood, which greatly simplifies the model.

We employ Collapsed Gibbs Sampling (CGS) [Neal, 2000] to estimate posterior over DP-GMM parameters given $\mathcal{D}$. CGS samples from the posterior by iteratively assigning observations to components. That is, given an observation $\boldsymbol{x}_i$, CGS samples a component $c_i$ from $p(c_i \mid \boldsymbol{c}_{-i}, \boldsymbol{x}_i, \alpha, \boldsymbol{\theta})$, where $\boldsymbol{\theta}$ are the parameters of the NIW posterior distributions over means and covariances. However, thanks to the conjugate prior on the component parameters, $p(c_i \mid \boldsymbol{c}_{-i}, \boldsymbol{x}_i, \alpha, \boldsymbol{\theta})$ does not depend on $\boldsymbol{\mu}_k$ and $\boldsymbol{\Sigma}_k$, as they can be marginalized out. This marginalization greatly reduces sampling variance [Liu et al., 1994]. That said, parameters for a given component can be easily recovered by sampling from the NIW posterior (see Appendix A for details):

$$p(\boldsymbol{\mu}_k, \boldsymbol{\Sigma}_k \mid \mathcal{D}, \boldsymbol{c}) = NIW(\boldsymbol{\mu}_k, \boldsymbol{\Sigma}_k \mid \boldsymbol{\theta}_k), \quad \boldsymbol{\theta}_k = \{\mathbf{m}_k, \nu_k, \kappa_k, \boldsymbol{S}_k\}. \tag{5}$$

An outcome of one CGS iteration is an assignment of observations to components. Collectively these iterations form a Markov chain that approximates the posterior distribution over DP-GMM parameters. In turn, this posterior induces a posterior predictive distribution for previously unseen observations: $p(\boldsymbol{x}^* \mid \mathcal{D})$, which can be seen as the model's view of the underlying data distribution. The posterior predictive given specific component assignments $\boldsymbol{c}_t$ (i.e. given a specific Gibbs sampling step):

$$p(\boldsymbol{x}^* \mid \mathcal{D}, \boldsymbol{c_t}) = \int p(\boldsymbol{x}^* \mid \boldsymbol{\mu}, \boldsymbol{\Sigma}, \boldsymbol{c_t}) p(\boldsymbol{\mu}, \boldsymbol{\Sigma} \mid \mathcal{D}, \boldsymbol{c_t}) d\boldsymbol{\mu} d\boldsymbol{\Sigma}$$

has a closed form solution (see Appendix B for details). The posterior predictive $p(\boldsymbol{x}^* \mid \mathcal{D})$ is an expectation over component assignments and can approximated by sampling steps from the Markov chain:

$$p(\boldsymbol{x}^* \mid \mathcal{D}) = \int p(\boldsymbol{x}^* \mid \mathcal{D}, \boldsymbol{c}) p(\boldsymbol{c}) d\boldsymbol{c} \approx \frac{1}{T} \sum_{t=1}^{T} p(\boldsymbol{x}^* \mid \mathcal{D}, \boldsymbol{c}_t). \tag{6}$$

## 3 Quantifying complexity and structure of posterior distributions

We use DP-GMM posterior predictive distributions to compare neural networks with respect to their representational complexity. To this end, we approximate a relative entropy between the posterior predictive $p(\boldsymbol{x}^* \mid \mathcal{D})$ and a chosen *least assumption* distribution $m(\boldsymbol{x}^*)$, i.e. the Kullback-Leibler (KL) divergence $D_{\mathrm{KL}}(p \,\|\, m)$. From an information theory point of view, this relative entropy can be seen as a measure of inefficiency of approximating the posterior predictive with $m(\boldsymbol{x}^*)$. Alternatively, $D_{\mathrm{KL}}(p \,\|\, m)$ can be seen as an information gain from observing many samples from $p(\boldsymbol{x}^* \mid \mathcal{D})$ while assuming $m(\boldsymbol{x}^*)$ prior. The measure obviously depends on the choice of $m(\boldsymbol{x}^*)$. We pick $m(\boldsymbol{x}^*)$ to be the maximum differential entropy distribution that captures mean of the data and variance in each dimension. That is, we choose the least assumption distribution to be a multivariate Gaussian with the mean and the diagonal covariance matrix estimated from $\mathcal{D}$.[3]

We do not have a closed-form expression for the relative entropy $D_{\mathrm{KL}}(p \,\|\, m)$. Fortunately, we can easily draw samples from the posterior predictive $p(\boldsymbol{x}^* \mid \mathcal{D})$ by first sampling a step from the CGS chain and then sampling from the posterior predictive given the component assignment (Eqn. 6). This gives us a Monte Carlo approximation to the relative entropy:

$$D_{\mathrm{KL}}(p \,\|\, m) \approx \frac{1}{TS} \sum_{t=1}^{T} \sum_{s=1}^{S} \left[ \log p(\boldsymbol{x}^*_{st} \mid \mathcal{D}, \boldsymbol{c}_t) - \log m(\boldsymbol{x}^*_{st}) \right], \quad \boldsymbol{x}^*_{st} \sim p(\boldsymbol{x}^* \mid \mathcal{D}, \boldsymbol{c}_t). \tag{7}$$

When modelling aggregated posteriors in VAEs we are also interested to what extent dimensions in the latent code are independent. To gauge the degree of dependency between latent dimensions, we estimate the total correlation between dimensions in posterior predictive. That is, we approximate the KL divergence between the full posterior predictive $p(\boldsymbol{z}^* \mid \mathcal{D})$ and its dimensions-independent version:

$$p_{ind}(\boldsymbol{z}^* \mid \mathcal{D}) = \prod_{i=1}^{D} p(z_i^* \mid \mathcal{D}). \tag{8}$$

Note that $p_{ind}(\boldsymbol{z}^* \mid \mathcal{D})$ is simply a product of marginals distribution. Again, KL divergence between $p$ and $p_{ind}$ has no closed-form solution. However, note that posterior predictive density $p(\boldsymbol{z}^* \mid \mathcal{D}, \boldsymbol{c})$ is a mixture of Student's t-distributions. Because marginals of a Student's t-distribution are also Student's t-distributions, $p(z_i^* \mid \mathcal{D}, \boldsymbol{c})$ can be expressed as a simple mixture:

$$p(z_i^* \mid \mathcal{D}, \boldsymbol{c}) = \sum_{k=1}^{K} \alpha_k \int St\left(\boldsymbol{z}^* \mid \boldsymbol{\mu}_k, \boldsymbol{\Sigma}_k, \nu_k\right) d\boldsymbol{z}^*_{-i} = \sum_{k=1}^{K} \alpha_k St\left(z_i^* \mid \mu_k^i, \Sigma_k^{ii}, \nu_k\right). \tag{9}$$

We can leverage this density to approximate $D_{\mathrm{KL}}(p \,\|\, p_{ind})$ with samples from the Markov chain:

$$D_{\mathrm{KL}}(p \,\|\, p_{ind}) \approx \frac{1}{TS} \sum_{t=1}^{T} \sum_{s=1}^{S} \left[ \log p(\boldsymbol{z}^*_{st} \mid \mathcal{D}, \boldsymbol{c}_t) - \log p_{ind}(\boldsymbol{z}^*_{st} \mid \mathcal{D}, \boldsymbol{c}_t) \right], \quad \boldsymbol{z}^*_{st} \sim p(\boldsymbol{z}^* \mid \mathcal{D}, \boldsymbol{c}_t).$$

$$\tag{10}$$

To approximate divergences in Eqn. 7 and 10 we perform $2,000$ CGS steps. Next, we throw away the first $1,000$ steps and thin the remaining part of the chain by taking every 20-th Gibbs step. We then calculate mean, minimum and maximum KL divergence across remaining Gibbs steps. In each step we take $10^5$ samples from the posterior predictive.

## 4 Representational complexity in Convolutional Networks

**Experimental setup.** First, we employ DP-GMMs to investigate representational complexity in CNNs that can exploit patterns in data and networks that are forced to memorize random labels. We also compare models with different depths, widths and regularization techniques. To this end, we train several CNN architectures on CIFAR-10 and Mini-ImageNet datasets[4]. Each network is trained with ground-truth labels and with a variant of the dataset in which labels were randomly permuted (further referred to as memorizing nets). All memorizing nets are trained on the same fixed random permutation of labels. Furthermore, when fitting true labels we train networks with no additional regularization, with image augmentation, with dropout and with both regularizers. See Appendix C for details on the datasets, network architectures and training hyper-parameters.

For each combination of a CNN architecture, label set and regularization, we train 50 networks starting from different random initializations and pool together their kernel representations (Section 2). One important choice when constructing CNN representations is the number of input images used to calculate kernel responses (Eqn. 2). On one hand, vector of kernel activations should form a distinct fingerprint of the learned feature. On the other hand, difficulty of estimating DP-GMM parameters increases with the dimensionality of representations. In practice we first collect kernel responses over the entire test part of the dataset. Assuming $l$ test images, a layer with $m$ kernels and $n$ trained networks, we obtain an $\mathbf{A}_{n \cdot m \times l}$ matrix with kernel representations. We then reduce the dimensionality of representations ($l$) by performing a Singular Value Decomposition of $\mathbf{A}$ and keeping only $d$ right-singular vectors with the largest singular values. We found that retaining up to $80$ singular vectors is sufficient to uncover differences in posterior distributions of kernel representations. We retain equal number of singular vectors when comparing layers trained under different scenarios.

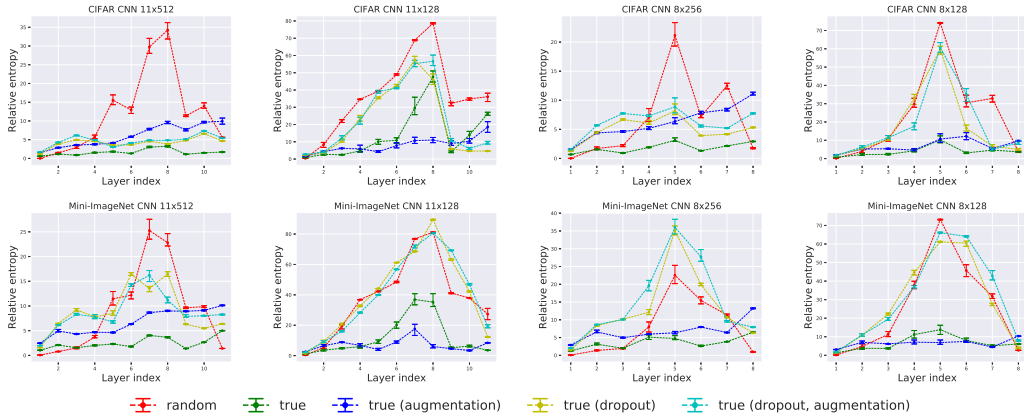

Figure 1: Relative entropies of posterior predictive distributions for CNN representations. Results are reported for true and randomly permuted labels, including dropout and image augmentation in the former case. In each case we report mean, minimum and maximum relative entropy across averaged Gibbs steps. In plot title, `CNN AxB` refers to a CNN with depth `A` and width `B` in the final conv layer.

**Representational complexity.** Results from CNN experiments are collected in Fig. 1. Additional results are reported in Appendix C. First, we observe that networks than can exploit patterns in data learn vastly less complex representations than networks forced to memorize, even though in principle both are perfectly capable of memorizing training examples [Zhang et al., 2017]. This finding supports conclusions drawn in [Arpit et al., 2017]. However, we also observe large differences in

effects of dropout compared to image augmentation or no regularization: dropout typically increases representational complexity. The extent of this increase depends on the network width, with narrow dropout nets learning representations with complexity more akin to that of memorizing nets. Dropout experiments also illustrate that low representational complexity is not a necessary prerequisite for generalization: while representations in dropout nets are highly sensitive to network initialization, they still form solutions that generalize. Finally, we observe increased representational complexity in middle layers of deep but narrow nets, when trained with no regularization (CNN 11x128 and CNN 11x192 in Appendix C). This is remedied by image augmentation, which behaves consistently across evaluated architectures.

## 5  Latent space structure in variational autoencoders

Variational autoencoders learn a variational posterior (or inference) distribution $q_\phi(z \mid x)$ and a generative distribution $p_\theta(x \mid z)$, by maximizing:

$$\mathcal{L}_\beta(\boldsymbol{x}, \boldsymbol{\theta}, \boldsymbol{\phi}) = \mathbb{E}_{q_\phi(\boldsymbol{z}|\boldsymbol{x})}\left[\log p_\theta(\boldsymbol{x} \mid \boldsymbol{z})\right] - \beta f(q_\phi(\boldsymbol{z} \mid \boldsymbol{x}), p(\boldsymbol{z})) \tag{11}$$

under a suitable divergence measure $f(q, p)$ between the posterior $q$ and prior $p$. In the standard VAE model $f(q, p)$ corresponds to the KL divergence, and $\beta = 1$. In such settings objective in Eqn. 11 is equivalent to the evidence lower bound on intractable data likelihood [Kingma and Welling, 2014]. Recently, however, there is an increasing interest in alternative formulations. Higgins et al. [2017] and Burgess et al. [2018] investigated VAEs with $\beta > 1$ and observed that such $\beta$-VAEs tend to learn disentangled latent codes $z$, i.e. codes where individual dimensions capture semantically meaningful properties of observations. Chen et al. [2017] suggests that $D_{\mathrm{KL}}$ can be a too restrictive regularization and may cause the model to learn uninformative latent codes. Zhao et al. [2017, 2019] studied VAEs with an alternative regularization, namely Maximum Mean Discrepancy (MMD) divergence [Gretton et al., 2012]. MMD was also investigated by Tolstikhin et al. [2018] in the context of Wasserstein autoencoders. Shortly, given a positive-definite kernel $k : \mathcal{Z} \times \mathcal{Z} \to \mathbb{R}$, MMD between two probability distributions $P$ and $Q$ on $\mathcal{Z}$ is a distance between their kernel mean embeddings. MMD has an unbiased estimator [Gretton et al., 2012] that easily integrates with gradient-based training:

$$MMD_k(P_Z, Q_Z) \approx \frac{1}{n(n-1)} \sum_{\substack{i,j=1 \\ i \neq j}}^{n} \left[k(z_i^p, z_j^p) + k(z_i^q, z_j^q)\right] - \frac{2}{n^2} \sum_{i,j=1}^{n} k(z_i^p, z_j^q), \tag{12}$$

where $\{z_i^p\}_{i=1}^n$, $\{z_i^q\}_{i=1}^n$ are samples from $P$ and $Q$, respectively.

In this section we leverage DP-GMMs to investigate aggregated posteriors (Eqn. 1) learned by VAEs across a range of $\beta$ values for both standard $D_{\mathrm{KL}}$ and MMD regularizations. This gives us a view into the structure of the latent spaces in these models. Additional results are reported in Appendix D.

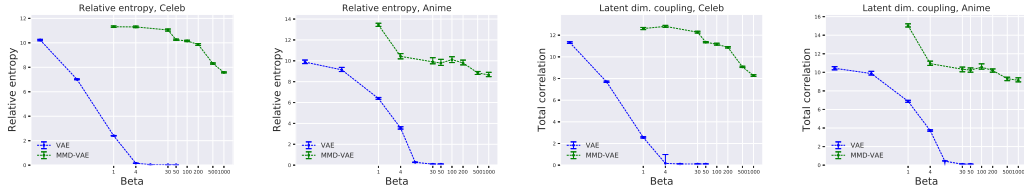

Figure 2: Relative entropies (left) and total correlations (right) in posterior predictive distributions for latent codes in $\beta$-VAEs and MMD-VAEs across a range of $\beta$ values. In each case we report mean, minimum and maximum estimate across averaged Gibbs steps.

**Experimental setup.**  All experiments were carried out on CelebA [Liu et al., 2015] and Anime[5] datasets consisting of images of human and animated character faces, respectively. Training protocols and network architectures follow those in Tolstikhin et al. [2018], particularly we learn latent codes with $d = 64$ dimensions and use inverse multiquadrics kernel in MMD-VAEs. See Appendix D for more details on dataset preparation and training hyper-parameters.

| $\beta$-VAE | | MMD-VAE | |
| --- | --- | --- | --- |
| $\beta = 0.01$ | $\beta = 30.$ | $\beta = 1.$ | $\beta = 1000.$ |

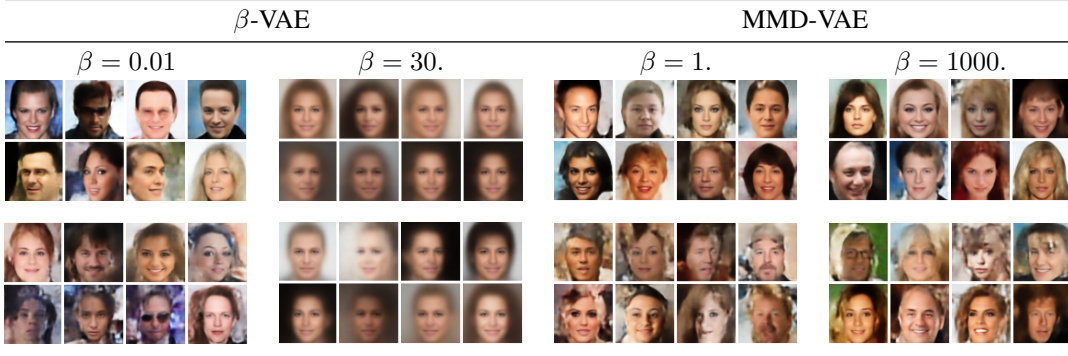

Figure 3: Samples generated with latent codes drawn from either the joint predictive density $p(\boldsymbol{z^*})$ (top) or product of marginals density $p_{ind}(\boldsymbol{z^*})$ (bottom) for VAEs trained on CelebA dataset.

After training a given model, we sample latent codes for the entire test part of the respective dataset and estimate a DP-GMM model for the set of sampled codes: $\mathcal{D}_z = \{\boldsymbol{z}_1, \boldsymbol{z}_2, \ldots, \boldsymbol{z}_n\}$. This gives us a CGS trace from which we can recover the posterior predictive $p(\boldsymbol{z^*} \mid \mathcal{D}_z)$ over the latent space learned by this particular VAE. We use this inferred distributions as proxies to investigate aggregated posteriors. For notational simplicity we will drop conditioning on $\mathcal{D}_z$ in the analysis below, and simply write $p(\boldsymbol{z^*})$ for the DP-GMM posterior predictive.

## 5.1   Latent space learned by $\beta$-VAEs and MMD-VAEs

We explore latent representations learned by VAEs in two ways. First, we quantify complexity of learned representations via relative entropies in posterior predictive distributions (Eqn. 7). Next, in order to investigate the degree of dependency between latent dimensions, we approximate total correlations between dimensions in posterior predictive densities (Eqn. 10).

**Effects of $\beta$ regularization on the aggregated posterior.**   Fig. 2 (left) shows relationship between $\beta$ value and the complexity of latent representations learned by standard $\beta$-VAEs and MMD-VAEs. This result demonstrates that $\beta$ has particularly strong regularizing effect on the aggregated posterior in standard $\beta$-VAEs: distribution of latent codes in this model rapidly simplifies as $\beta$ coefficient grows. For $\beta > 10$, aggregated posterior becomes almost indistinguishable from a diagonal multivariate normal distribution with mean and variance estimated from $\mathcal{D}_z$ (i.e. the lest-assumption distribution in the construction of relative entropy). In other words, posterior in $\beta$-VAEs with strong regularization collapses to the prior. Regularization is much weaker under MMD divergence, where relative entropies indicate rich latent space even with large $\beta$ values ($\beta = 1000$).

**Independence of latent dimensions.**   $\beta$-VAEs were observed to learn disentangled representations when trained with large $\beta$ values [Higgins et al., 2017]. Here we leverage posterior predictive $p(\boldsymbol{z^*})$ to investigate influence of large $\beta$ on the covariance structure of the aggregated posterior $q_\phi(\boldsymbol{z})$. Fig. 2 (right) demonstrates that latent dimensions in standard $\beta$-VAEs decorrelate with increasing $\beta$ value: joint predictive density over latent codes becomes indistinguishable from its product of marginals approximation. This agrees with the disentanglement phenomenon observed in these models. MMD-VAEs, on the other, hand keep their latent codes relatively correlated, even with strong regularization.

To further illustrate how $\beta$ regularization affects coupling between latent dimensions, we also sampled VAE observations with latent codes drawn either from a joint posterior predictive $p(\boldsymbol{z^*})$ or a product of marginals density $p_{ind}(\boldsymbol{z^*})$ (Eqn. 8). Samples from MMD-VAEs and standard $\beta$-VAEs trained with small $\beta$ often degrade when dependence between latent dimensions is dropped (Fig. 3). In a strongly regularized $\beta$-VAE samples from the joint and the product of marginals distributions are indistinguishable, but a simplistic latent space translates to low sample fidelity and diversity. Overall, our results show that disentanglement in $\beta$-VAEs comes at the cost of reduced representational capacity.

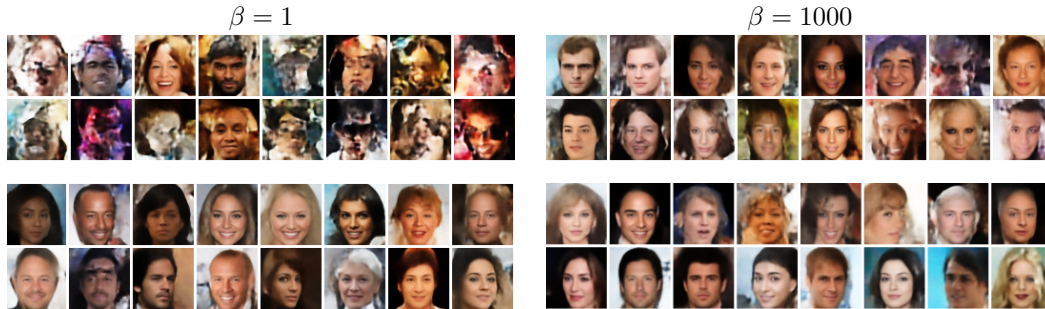

Figure 4: MMD-VAE samples generated with latent codes drawn from either the prior $p(\boldsymbol{z})$ (top) or DP-GMM posterior predictive $p(\boldsymbol{z}^* \mid \boldsymbol{c})$ (bottom). Results for models trained on CelebA dataset.

## 5.2 Improving samples from VAEs with rich posteriors

Results presented above show that aggregated posteriors in MMD-VAEs diverge significantly from the prior. This suggests that sampling in MMD-VAEs can be improved by drawing latent codes from an approximation to $q_\phi(\boldsymbol{z})$, rather than the prior $p(\boldsymbol{z})$. In fact, posterior predictive given component assignments $p(\boldsymbol{z}^* \mid \boldsymbol{c})$ is a natural choice for such approximation. First, it admits an efficient ancestral sampling, where we first sample a component and then the latent code. Second, given flexibility of DP-GMMs, we may expect that after initial burn-in period mixtures in the chain will be well adapted to $q_\phi(\boldsymbol{z})$. Figure 4 compares this sampling scheme with a standard sampling from the prior. Clearly, sampling latent codes from a mixture $p(\boldsymbol{z}^* \mid \boldsymbol{c})$ significantly improves quality of image samples. Note also that large $\beta$ term only partially remedies issues with samples generated from the prior. We could also sample from the full posterior predictive by first sampling a step from the Markov chain. This could further improve sample diversity at the cost of storing more posterior parameters.

## 6 Related work

Several recent works explored similarity of representations learned by neural networks. Raghu et al. [2017] construct neurons' representations as vectors of their responses over a fixed set of inputs. This differs from a typical notion of a neural representation understood as a vector of activations in a network layer given a single input example. They show that representations learned by networks trained from different initializations exhibit similarity in canonical directions. A follow up work by Morcos et al. [2018] proposes an alternative way to subsume correlation in canonical directions. They study similarity of neural representations in memorizing and learning networks, compare similarity of representations in wide and narrow networks and investigate training dynamics in RNNs. More recently, Kornblith et al. [2019] proposed a kernel-based similarity index that more reliably captures correspondence between network representations. This allowed them, among others, to pinpoint depth-related pathologies in convolutional networks. The main difference between these works and our approach is that we do not seek to construct a similarity score for pairs of layer representations. We instead investigate distributions of neural representations learned across many trained networks and study aggregated posteriors in deep generative models. Rather than focusing mainly on network similarity, our goal is to compare networks with respect to the complexity of effectively learnable representations or structure of the learned latent space. This requires a more flexible tool than a similarity score, which in our case is a nonparametric mixture model. A work more akin to ours was presented by Montavon et al. [2011], whose aim was to verify whether successive network layers construct representations that are increasingly good at solving the underlying task. Still, their analysis sheds no light on the complexity of the set of representations that can be effectively reached by a specific network architecture and training regime.

Our work also touches on the effects of memorization on learned representations. Zhang et al. [2017] demonstrate that neural networks easily memorize random assignment of labels and random input examples. An immediate conclusion from this work is that priors encoded in current network architectures are not a factor that could prevent memorization. If so, then is the observed efficacy of neural networks actually due to learning patterns in data? Arpit et al. [2017] compare how memorizing networks and networks trained on real data fit input examples. They demonstrate that the latter fit

simple examples first. They also show that memorizing networks have more complex decision boundaries. Wilson and Izmailov [2020] demonstrate that memorization of images with random labels can be replicated with Gaussian processes. They then discuss generalization from a perspective of priors over functions that are encoded by composing model architectures with priors over their parameters. They argue that for CNNs these prior distributions concentrate on functions that exploit patterns in data, and attribute memorization to non-zero prior density for random label assignments. In particular, they demonstrate that a simple CNN with random weights induce a covariance structure in MNIST images that correlates with ground-truth labels. We contribute to this line of research by demonstrating that the set of representations that are effectively constructed by memorizing networks is more complex than the set of representations constructed by networks that learn on true data. This shows that CNNs that can exploit patterns in data converge do different solutions than memorizing nets, despite no difference in architecture, regularization or training hyper-parameters.

Our results demonstrate that disentanglement in standard $\beta$-VAEs comes with a simplistic aggregated posterior, which translates to reduced fidelity and diversity of samples. Gao et al. [2019] investigate learning of disentangled representations in a Correlation Explanation (CorEx) framework [Steeg and Galstyan, 2014]. Their basic idea is to learn a parametrized probability distribution $p_{\boldsymbol{\theta}}(\mathbf{x}, \mathbf{z})$ which jointly maximizes the total correlation in $\mathbf{x}$ that is explained by the latent code $\mathbf{z}$ and minimizes total correlation in the latent code itself. Gao et al. formulate a variational lower bound to CorEx and show that under certain assumptions it is equivalent to ELBO in VAEs. From this perspective, $\beta$ regularization controls the contribution of mutual information between observations and latent dimensions to the optimization objective. Gao et al. also propose to improve samples in their model by drawing latent codes from a factorial approximation to the aggregated posterior. Our empirical results for standard $\beta$-VAEs are compatible with Gao et al. findings. That said, our framework can also be used to investigate aggregated posteriors in VAEs with non-standard divergences, such as MMD-VAEs. In these models a factorial approximation to the aggregated posterior yields poor samples, which we remedy by approximating the posterior with a Gaussian mixture.

While in this work we compare distributions of neural representations via relative entropies, one could argue that the number of components in a posterior distribution is itself a useful proxy to representational complexity. For example, sample complexity of learning a Gaussian mixture is linear (up to a poly-logarithmic factor) in the number of components [Ashtiani et al., 2018]. Note, however, that Dirichlet Process prior is not a suitable tool for recovering component counts in mixture distributions. Dirichlet Process is a prior on infinite mixtures and will not concentrate on a finite number of components in the infinite data limit [Miller and Harrison, 2013, 2014]. One can obtain consistency for the number of components with a suitable prior on finite mixtures [Miller and Harrison, 2018]. Still, analysis of component counts comes with caveats. It assumes that observations actually come from a finite mixture and that the form of the components' distribution is know – a fairly strong assumptions for a complex generative process behind neural representations. For these reasons we draw our conclusions from predictive densities, not component counts.

## 7   Conclusions

We presented a Bayesian Nonparametrics framework for investigating neural representations. The main strength of this probabilistic approach is that it allows us to investigate representations that are effectively reachable by gradient-based training, rather than quantifying only the theoretical model complexity. We used it to compare complexity of representations learned by CNNs and to explore structure of latent spaces learned by VAEs. Our results show marked differences between memorizing networks and networks that learn on true data, as well as between two form of regularization, namely dropout and image augmentation. Finally, we showed marked differences between standard $\beta$-VAEs and MMD-VAEs with respect to their ability to represent diverse image features in the latent space.

Our complexity analysis may have direct applications in development of latent variable generative models. First, it enables model comparison with respect to the capacity of the learned latent space. Second, we show that Gaussian mixtures can be used to improve samples from models with rich posteriors. Our results may also have immediate applications in interpretability research. A number of interpretation methods attempt explanation by capturing semantics of network units [Gilpin et al., 2018]. However, we uncover cases, such as dropout nets, where learned representations are sensitive to network initialization, raising doubts whether capturing semantics of network units is useful in these settings.

## 8 Broader Impact

This work have direct applications in deep generative models. Probabilistic models of latent spaces may inform development of architectures and training methods that improve sample fidelity and control over sample semantics. While generative modelling have many positive applications – e.g. in computer aided art and conversational systems – any work on generative models may potentially be used to produce deceptive and fraudulent content. This work also adds to the evidence that convolutional networks excel at exploiting patterns in data. However, it is important to recognize that our results do not speak to the issue of biases that may be inherited from training examples. In particular, undue trust in data-driven systems – including neural networks – runs the risk of reinforcing biases and prejudice existing in training data.

## Acknowledgements

Research presented in this work was supported by funds assigned to AGH University of Science and Technology by the Polish Ministry of Science and Higher Education. This research was supported in part by PL-Grid Infrastructure.

## Footnotes

[1]Separately for each layer and using same sequence of input images in each case.

[2]Arpit et al. [2017] define effective capacity of a learning algorithm as a set of all hypotheses that can be effectively constructed by that algorithm. This definition considers a hypothesis effectively learnable if there exists a dataset on which it is learned by that algorithm. Obviously, we quantify complexity of learned representations on some chosen but representative learning tasks.

[3]Both are maximum likelihood estimates.

[4]While Mini-ImageNet is typically used for few-shot learning, in this work we use the provided labels to train plain image classification nets.

[5]`https://github.com/Mckinsey666/Anime-Face-Dataset`

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
