[Supplementary Material]

# A Bayesian Nonparametrics View into Deep Representations

# Supplementary material

## A   Collapsed Gibbs Sampling for DP-GMM

We estimate posterior distributions over DP-GMM parameters using Collapsed Gibbs Sampling (CGS) algorithm [Neal, 2000]. Here we describe CGS in more details. The derivation below closely follows the one from [Murphy, 2012, chapters 24.2.4 & 25.2.4].

CGS steps consist of iterative sampling of component indicators $c_i$ for observations $\boldsymbol{x}_i \in \mathcal{D}$, $i = 1, \ldots, N$. During this sampling, CGS maintains a set of $K$ components, where $K$ itself is a random variable (components can be added or removed). Each component $k \in \{1, \ldots, K\}$ is described by a set $\boldsymbol{\theta}_k$ of parameters of the Normal-inverse-Wishart posterior distribution over its mean $\boldsymbol{\mu}_k$ and covariance $\boldsymbol{\Sigma}_k$:

$$p(\boldsymbol{\mu}_k, \boldsymbol{\Sigma}_k \mid \mathcal{D}) = p(\boldsymbol{\mu}_k, \boldsymbol{\Sigma}_k \mid \boldsymbol{\theta}_k) = NIW(\boldsymbol{\mu}_k, \boldsymbol{\Sigma}_k \mid \boldsymbol{\theta}_k), \quad \boldsymbol{\theta}_k = \{\boldsymbol{m}_k, \nu_k, \kappa_k, \boldsymbol{S}_k\}. \quad (1)$$

Given a vector of current component assignments $\boldsymbol{c} = \{c_1, \ldots, c_N\}$, CGS samples a new assignment for $\boldsymbol{x}_i$ from the conditional $p(c_i \mid \boldsymbol{c}_{-i}, \boldsymbol{x}_i, \alpha, \boldsymbol{\theta})$:

$$p(c_i = k \mid \boldsymbol{c}_{-i}, \boldsymbol{x}_i, \alpha, \boldsymbol{\theta}) \propto p(c_i = k \mid \boldsymbol{c}_{-i}, \alpha) p(\boldsymbol{x}_i \mid \boldsymbol{x}_{-i}, c_i = k, \boldsymbol{c}_{-i}, \boldsymbol{\theta}). \quad (2)$$

Here $\boldsymbol{c}_{-i}, \boldsymbol{x}_{-i}$ stand for, respectively, component assignments and the dataset with the $i$-th element removed; $\boldsymbol{\theta}$ is the collection of all component parameters. The first term in the above product comes from the Chinese-Restaurant Process (CRP), and has the form:

$$p(c_i = k \mid \boldsymbol{c}_{-i}, \alpha) = \begin{cases} \frac{N_{k,-i}}{\alpha+N-1} & \text{if } k \text{ already exists}, \\ \frac{\alpha}{\alpha+N-1} & \text{if } k \text{ is a new component}. \end{cases} \quad (3)$$

The second term can be derived by noticing that given the component assignment ($c_i = k$), observation $\boldsymbol{x}_i$ becomes conditionally independent from observations assigned to other components:

$$p(\boldsymbol{x}_i \mid \boldsymbol{x}_{-i}, c_i = k, \boldsymbol{c}_{-i}, \boldsymbol{\theta}) = p(\boldsymbol{x}_i \mid \boldsymbol{x}_{-i,k}, \boldsymbol{\theta}_k), \quad (4)$$

where $\boldsymbol{x}_{-i,k}$ are observations assigned to the $k$-th component, except $\boldsymbol{x}_i$. The above term has a closed-form solution computed by integrating out $\boldsymbol{\mu}_k, \boldsymbol{\Sigma}_k$ parameters, namely a multivariate Student's t-distribution (more details are given in appendix B):

$$p(\boldsymbol{x}_i \mid \boldsymbol{x}_{-i,k}, \boldsymbol{\theta}_k) = p(\boldsymbol{x}_i \mid \boldsymbol{\theta}_k) = \int p(\boldsymbol{x}_i \mid \boldsymbol{\mu}_k, \boldsymbol{\Sigma}_k) p(\boldsymbol{\mu}_k, \boldsymbol{\Sigma}_k \mid \boldsymbol{\theta}_k) d\boldsymbol{\mu}_k d\boldsymbol{\Sigma}_k. \quad (5)$$

When $k$ is a new (empty) component, the above density reduces to the Student's t-distribution with parameters derived from the NIW prior hyper-parameters, i.e. from $\boldsymbol{\theta}_0 = \{\boldsymbol{m}_0, \nu_0, \kappa_0, \boldsymbol{S}_0\}$:

$$p(\boldsymbol{x}_i \mid \boldsymbol{\theta}_0) = \int p(\boldsymbol{x}_i \mid \boldsymbol{\mu}_k, \boldsymbol{\Sigma}_k) p(\boldsymbol{\mu}_k, \boldsymbol{\Sigma}_k \mid \boldsymbol{\theta}_0). \quad (6)$$

Note also that the density from which we are sampling component assignments $p(c_i \mid \boldsymbol{c}_{-i}, \boldsymbol{x}_i, \alpha, \boldsymbol{\theta})$ is a categorical distribution over $c_i$. After sampling from it, one has to update the parameters of the NIW posterior of the newly sampled component (to account for the new member $\boldsymbol{x}_i$). Further, before calculating any quantities necessary to compute $p(c_i \mid \boldsymbol{c}_{-i}, \boldsymbol{x}_i, \alpha, \boldsymbol{\theta})$ (e.g. predictive densities for $\boldsymbol{x}_i$), observation $\boldsymbol{x}_i$ must be removed from the old component, reflecting the conditioning on $\boldsymbol{x}_{-i,k}$.

The probability distribution for component assignments (Eqn. 2) depends on $\alpha$, i.e. the concentration parameter in the Dirichlet Process prior. We do not assume a fixed value for $\alpha$, but explicitly account

for its uncertainty by treating it as a model parameter. To this end, we put a $Gamma(1,1)$ prior on $\alpha$. Under this prior the conditional $p(\alpha \mid K, N)$ – which is required by CGS – admits a simple sampling [Escobar and West, 1995].

We choose the values of hyper-parameters $\boldsymbol{\theta}_0$ in a way that puts a weakly-informative, data-dependent prior on component means and covariances. Specifically, we follow guidelines in [Fraley and Raftery, 2007], and set the $\boldsymbol{\theta}_0$ values to:

$$\boldsymbol{m}_0 = \frac{1}{N} \sum_{i=1}^{N} \boldsymbol{x}_i, \ \ \boldsymbol{S}_0 = \frac{\text{diag}(\boldsymbol{S}_{\overline{x}})}{K_0^{2/D}}, \tag{7}$$

$$\nu_0 = D + 2, \ \ \kappa_0 = 0.01,$$

where $D$ is the data dimensionality, and:

$$K_0 = \alpha \log\left(1 + \frac{N}{\alpha}\right), \ \ \boldsymbol{S}_{\overline{x}} = \frac{1}{N} \sum_{i=1}^{N} (\boldsymbol{x}_i - \boldsymbol{m}_0)(\boldsymbol{x}_i - \boldsymbol{m}_0)^T. \tag{8}$$

Here, $K_0$ is the prior expectation for the number of components and $\boldsymbol{S}_{\overline{x}}$ is the empirical covariance.

## B  DP-GMM posterior predictive distribution

Here we derive the formula for the Posterior Predictive distribution given component assignments: $p(\boldsymbol{x}^* \mid \mathcal{D}, \boldsymbol{c})$. The output from CGS is a Markov chain, where at each step we have component assignments $\boldsymbol{c} = \{c_i\}_{i=1}^{N}$ and posterior distributions $p(\boldsymbol{\mu}_k, \boldsymbol{\Sigma}_k \mid \mathcal{D})$ over component parameters $\{\boldsymbol{\mu}_k, \boldsymbol{\Sigma}_k\}_{k=1}^{K}$. Because of the conjugate NIW prior choice for means and covariances, the posterior is also a NIW distribution:

$$p(\boldsymbol{\mu}_k, \boldsymbol{\Sigma}_k) = NIW(\boldsymbol{\mu}_k, \boldsymbol{\Sigma}_k \mid \mathbf{m}_k, \nu_k, \kappa_k, \boldsymbol{S}_k). \tag{9}$$

The posterior predictive over a new observation $\boldsymbol{x}^*$ given the dataset $\mathcal{D}$ and component assignments $\boldsymbol{c}$ can be written as:

$$p(\boldsymbol{x}^* \mid \mathcal{D}, \boldsymbol{c}) = \int p(\boldsymbol{x}^* \mid \boldsymbol{\theta}, \boldsymbol{c}) p(\boldsymbol{\theta} \mid \mathcal{D}, \boldsymbol{c}) d\boldsymbol{\theta}$$
$$= \int \left[\sum_{k=1}^{K} p(\boldsymbol{x}^* \mid \boldsymbol{\theta}_k) p(c^* = k \mid \boldsymbol{c})\right] p(\boldsymbol{\theta} \mid \mathcal{D}, \boldsymbol{c}) d\boldsymbol{\theta}. \tag{10}$$

Posterior over component parameters expands to:

$$p(\boldsymbol{\theta} \mid \mathcal{D}, \boldsymbol{c}) = p\left(\{\boldsymbol{\mu}_k, \boldsymbol{\Sigma}_k\}_{k=1}^{K} \mid \mathcal{D}, \boldsymbol{c}\right) = \prod_{k=1}^{K} p(\boldsymbol{\mu}_k, \boldsymbol{\Sigma}_k \mid \mathcal{D}_k). \tag{11}$$

Above, we leverage the fact that given component assignments the posterior over joint set of parameters factorizes over components; $\mathcal{D}_k$ denotes observations assigned to the $k$-th component, i.e $\mathcal{D}_k = \{\boldsymbol{x}_i : c_i = k\}$. From the model definition it is clear that:

$$p(\boldsymbol{x}^* \mid c^* = k) = p(\boldsymbol{x}^* \mid \boldsymbol{\mu}_k, \boldsymbol{\Sigma}_k). \tag{12}$$

Now, let $\alpha_k = p(c^* = k \mid \boldsymbol{c})$ be the predictive mixture weights. By plugging Eqns. 11 and 12 into Eqn. 10 we obtain:

$$\int \left[\sum_{k=1}^{K} \alpha_k p(\boldsymbol{x}^* \mid \boldsymbol{\mu}_k, \boldsymbol{\Sigma}_k)\right] \prod_{j=1}^{K} p(\boldsymbol{\mu}_j, \boldsymbol{\Sigma}_j \mid \mathcal{D}_j) d\boldsymbol{\mu} d\boldsymbol{\Sigma}$$

$$= \sum_{k=1}^{K} \int \alpha_k p(\boldsymbol{x}^* \mid \boldsymbol{\mu}_k, \boldsymbol{\Sigma}_k) \prod_{j=1}^{K} p(\boldsymbol{\mu}_j, \boldsymbol{\Sigma}_j \mid \mathcal{D}_j) d\boldsymbol{\mu} d\boldsymbol{\Sigma}$$

$$= \sum_{k=1}^{K} \int \left[\int \alpha_k p(\boldsymbol{x}^* \mid \boldsymbol{\mu}_k, \boldsymbol{\Sigma}_k) p(\boldsymbol{\mu}_k, \boldsymbol{\Sigma}_k \mid \mathcal{D}_k) d\boldsymbol{\mu}_k d\boldsymbol{\Sigma}_k\right] \prod_{j \neq k} p(\boldsymbol{\mu}_j, \boldsymbol{\Sigma}_j \mid \mathcal{D}_j) d\boldsymbol{\mu}_{-k} d\boldsymbol{\Sigma}_{-k}$$

$$= \sum_{k=1}^{K} \alpha_k \int p(\boldsymbol{x}^* \mid \boldsymbol{\mu}_k, \boldsymbol{\Sigma}_k) p(\boldsymbol{\mu}_k, \boldsymbol{\Sigma}_k \mid \mathcal{D}_k) d\boldsymbol{\mu}_k d\boldsymbol{\Sigma}_k,$$

$$\tag{13}$$

where $\boldsymbol{\mu}_{-k}$ and $\boldsymbol{\Sigma}_{-k}$ are jointly means and covariances of components other than the $k$-th one. Expression under the last integral in Eqn. 13 is tractable, thanks to the conjugacy of the Normal-inverse-Wishart prior to the Gaussian likelihood. Specifically, it is the probability density function of a multivariate Student's t-distribution Murphy [2012, 2007]:

$$\int \mathcal{N}(\boldsymbol{x}^* \mid \boldsymbol{\mu}_k, \boldsymbol{\Sigma}_k) NIW(\boldsymbol{\mu}_k, \boldsymbol{\Sigma}_k \mid \boldsymbol{m_k}, \nu_k, \kappa_k, \boldsymbol{S}_k) d\boldsymbol{\mu}_k d\boldsymbol{\Sigma}_k$$
$$= St\left(\boldsymbol{x}^* \mid \boldsymbol{m}_k, \frac{\kappa_k + 1}{\kappa_k(\nu_k - D + 1)} \boldsymbol{S}_k, \nu_k - D + 1\right). \tag{14}$$

Finally, posterior predictive density (10) can be written as a mixture of multivariate Student's t-distributions with weights $\{\alpha_{k=1}^K\}$:

$$p(\boldsymbol{x}^* \mid \mathcal{D}, \boldsymbol{c}) = \sum_{k=1}^K \alpha_k St\left(\boldsymbol{x}^* \mid \boldsymbol{m}_k, \beta_k \boldsymbol{S}_k, \nu_k - D + 1\right), \quad \beta_k = \frac{\kappa_k + 1}{\kappa_k(\nu_k - D + 1)}. \tag{15}$$

The only thing left is to derive the predictive mixture weights: $\{\alpha_{k=1}^K\}$. Recall that $\alpha_k$ is the probability of choosing the $k$-th component for a new observation given the existing component assignments $\boldsymbol{c}$. The set of predictive weights $\{\alpha_{k=1}^K\}$ therefore forms a categorical distribution $p(c^* \mid \boldsymbol{c})$ over the components. It can be calculated by marginalizing out the Dirichlet distributed component weights:

$$\alpha_k = p(c^* = k \mid \boldsymbol{c}) = \int p(c^* = k \mid \boldsymbol{\pi}) p(\boldsymbol{\pi} \mid \boldsymbol{c}) d\boldsymbol{\pi}$$
$$= \int \text{Cat}(c^* = k \mid \boldsymbol{\pi}) \text{Dir}(\boldsymbol{\pi} \mid \boldsymbol{c}) d\boldsymbol{\pi} = \frac{N_k + \alpha_0}{N(1 + \alpha_0)}, \tag{16}$$

where $\alpha_0$ is the concentration parameter in the Dirichlet Process prior and $N_k$ is the number of observations assigned to the $k$-th component. Note that predictive mixture weights depend only on the concentration parameter and the number of observations assigned to each component.

## C CNN experiment details and additional results

**Datasets and image augmentation.** CNN experiments (Section 4) were run on CIFAR-10 [Krizhevsky, 2009] and Mini-ImageNet [Vinyals et al., 2016] datasets. To adapt Mini-ImageNet to a typical image classification task, we concatenated the provided training, validation and test subsets (without support and evaluation splits in training data) and then randomly split the dataset into $50,000$ training and $10,000$ test examples. Finally, we resized the images to $42 \times 42$ pixels. CIFAR experiments used the standard train/test split. Image augmentation was implemented as a random $32 \times 32$ (CIFAR) or $42 \times 42$ (Mini-ImageNet) pixel crop from an input with 4 (CIFAR) or 5 (Mini-ImageNet) pixel padding, followed by a random horizontal flip.

**Network architectures and training hyper-parameters.** CNN architectures used in experiments are summarized in Table C.1. All networks were trained for 60 epochs using stochastic gradient descent with learning rate $\epsilon = 0.01$, momentum $\mu = 0.9$ (with Nesterov accelerated gradient), $L_2$ penalty $\lambda = 10^{-6}$ and a batch size equal to 512 examples. Networks with additional regularization had 20% dropout after each nonlinearity.

**Additional results.** Results for architectures not included in Section 4 are summarized in Fig. C.1. These results give further support to conclusions from main CNN experiments: a) memorizing networks learn more complex representations than networks trained on true labels, b) unlike image augmentation, dropout significantly increases representational complexity in CNNs; this effect diminishes with network width, and c) absent image augmentation, deep but narrow nets exhibit increased representational complexity in middle layers.

## D VAE experiment details and additional results

**Datasets.** VAE experiments were carried out on CelebA [Liu et al., 2015] and Anime [Ani] datasets. CelebA examples were prepared by taking a $150 \times 150$ central crop and resizing it to $64 \times 64$ pixels. Anime images were resized to $96 \times 96$ pixels. Both datasets were randomly split into train/test subsets: 150000/52599 split for CelebA and 50000/13632 split for Anime images.

| CNN 11x512 | CNN 11x384 | CNN 11x256 | CNN 11x192 | CNN 11x128 |
|---|---|---|---|---|
| conv. $3 \times 3$ $c = 64$ | conv. $3 \times 3$ $c = 48$ | conv. $3 \times 3$ $c = 32$ | conv. $3 \times 3$ $c = 24$ | conv. $3 \times 3$ $c = 16$ |
| same | | | | |
| conv. $3 \times 3$ $c = 128, s = 2$ | conv. $3 \times 3$ $c = 96, s = 2$ | conv. $3 \times 3$ $c = 64, s = 2$ | conv. $3 \times 3$ $c = 48, s = 2$ | conv. $3 \times 3$ $c = 32, s = 2$ |
| conv. $3 \times 3$ $c = 128$ | conv. $3 \times 3$ $c = 96$ | conv. $3 \times 3$ $c = 64$ | conv. $3 \times 3$ $c = 48$ | conv. $3 \times 3$ $c = 32$ |
| same | | | | |
| conv. $3 \times 3$ $c = 256, s = 2$ | conv. $3 \times 3$ $c = 192, s = 2$ | conv. $3 \times 3$ $c = 128, s = 2$ | conv. $3 \times 3$ $c = 96, s = 2$ | conv. $3 \times 3$ $c = 64, s = 2$ |
| conv. $3 \times 3$ $c = 256$ | conv. $3 \times 3$ $c = 192$ | conv. $3 \times 3$ $c = 128$ | conv. $3 \times 3$ $c = 96$ | conv. $3 \times 3$ $c = 64$ |
| same | | | | |
| conv. $3 \times 3$ $c = 512, s = 2$ | conv. $3 \times 3$ $c = 384, s = 2$ | conv. $3 \times 3$ $c = 256, s = 2$ | conv. $3 \times 3$ $c = 192, s = 2$ | conv. $3 \times 3$ $c = 128, s = 2$ |
| conv. $3 \times 3$ $c = 512$ | conv. $3 \times 3$ $c = 384$ | conv. $3 \times 3$ $c = 256$ | conv. $3 \times 3$ $c = 192$ | conv. $3 \times 3$ $c = 128$ |
| CIFAR: same; Mini-ImageNet: same with $s = 2$ | | | | |
| Fully Connected (logits) | | | | |

| CNN 8x256 | CNN 8x192 | CNN 8x128 |
|---|---|---|
| conv. $3 \times 3$ $c = 64$ | conv. $3 \times 3$ $c = 48$ | conv. $3 \times 3$ $c = 32$ |
| same | | |
| conv. $3 \times 3$ $c = 128, s = 2$ | conv. $3 \times 3$ $c = 96, s = 2$ | conv. $3 \times 3$ $c = 64, s = 2$ |
| conv. $3 \times 3$ $c = 128$ | conv. $3 \times 3$ $c = 96$ | conv. $3 \times 3$ $c = 64$ |
| same | | |
| conv. $3 \times 3$ $c = 256, s = 2$ | conv. $3 \times 3$ $c = 192, s = 2$ | conv. $3 \times 3$ $c = 128, s = 2$ |
| conv. $3 \times 3$ $c = 256$ | conv. $3 \times 3$ $c = 192$ | conv. $3 \times 3$ $c = 128$ |
| CIFAR: same; Mini-ImageNet: same with $s = 2$ | | |
| Fully Connected (logits) | | |

Table C.1: CNN architectures used in experiments (Section 4). Each convolutional layer is followed by batch normalization and ReLU nonlinearity. Dropout – when used – is applied after ReLU. All conv layers use 1 pixel zero-padding. $c$ - number of output channels; $s$ - layer stride (default 1).

**Model architectures and training details.** Architectures for encoder and decoder networks follow closely those used by Tolstikhin et al. [2018] and are summarized in the table D.2. All models had a latent space with $d = 64$ dimensions. Training was carried out for 60 epochs using Adam optimizer [Kingma and Ba, 2015] with a constant learning rate: $\epsilon = 0.001$ and a batch size equal to 64 images. We trained standard $\beta$-VAE models with $\beta \in \{0.01, 0.1, 1.0, 4.0, 10.0, 30.0, 50.0\}$ and MMD-VAE models with $\beta \in \{1.0, 4.0, 30.0, 50.0, 100.0, 200.0, 500.0, 1000.0\}$.

Figure C.1: Relative entropies of posterior predictive distributions for CNN representations. Results for CNN architectures not included in Section 4.

| Encoder | Decoder |
|---|---|
| conv. $5 \times 5$, $c = 128$, $s = 2$ | Fully Connected: $8 \cdot 8 \cdot 1024$ (Celeb) or $12 \cdot 12 \cdot 1024$ (Anime) |
| conv. $5 \times 5$, $c = 256$, $s = 2$ | FS conv. $5 \times 5$, $c = 512$, $s = 2$ |
| conv. $5 \times 5$, $c = 512$, $s = 2$ | FS conv. $5 \times 5$, $c = 256$, $s = 2$ |
| conv. $5 \times 5$, $c = 1024$, $s = 2$ | FS conv. $5 \times 5$, $c = 128$, $s = 2$ |
| Fully Connected: $64 \cdot 2$ | FS conv. $5 \times 5$, $c = 3$, $s = 1$ |

Table D.2: Convolutional encoder and decoder architectures used in VAE experiments. FS conv. – fractionally strided convolution; $c$ – number of output channels, $s$ – stride. Each convolution and fractionally strided convolution is followed by a batch normalization and ReLU nonlinearity.

**Component counts in CGS traces.** Our primary tools for judging complexity of aggregated posteriors in VAE models are relative entropy and total correlation between dimensions in posterior predictive (Section 3). That said, findings from Section 5 are further supported by distributions of component counts in CGS traces, pictured in Fig. D.2. In standard $\beta$-VAEs, increasing regularization strength simplifies the set of inferred latent codes to the point where it can be explained (by DP-GMM) using just one component. In MMD-VAEs regularization has no obvious influence on the number of components in CGS traces. Note, however, that these results should be interpreted with care – specifically, they do not speak to the number of components in the data generating distribution. Indeed, Dirichlet Process is not consistent for the number of components [Miller and Harrison, 2013, 2014]. Rather, results in Fig. D.2 can be seen as the number of components (from an infinite mixture) observed in a finite set of available data points.

Figure D.2: Component counts in CGS traces for standard $\beta$-VAEs and MMD-VAEs. In each case we report mean, minimum and maximum component count across sampled CGS steps.

**Additional samples.** Figures D.3 and D.4 present additional image samples generated with latent codes drawn from a posterior predictive density $p(z^* \mid c)$.

**CelebA samples**

| Standard $\beta$-VAE | MMD-VAE |
|---|---|
| $\beta = 0.01$ | $\beta = 1.$ |

| $\beta = 0.1$ | $\beta = 4.$ |
|---|---|

| $\beta = 1.$ | $\beta = 30.$ |
|---|---|

| $\beta = 4.$ | $\beta = 50.$ |
|---|---|

| $\beta = 10.$ | $\beta = 100.$ |
|---|---|

| $\beta = 30.$ | $\beta = 200.$ |
|---|---|

Figure D.3: CelebA samples generated with latent codes drawn from a posterior predictive density.

**Anime samples**

| Standard $\beta$-VAE | MMD-VAE |
|---|---|

$\beta = 0.01$ | $\beta = 1.$

$\beta = 0.1$ | $\beta = 4.$

$\beta = 1.$ | $\beta = 30.$

$\beta = 4.$ | $\beta = 50.$

$\beta = 10.$ | $\beta = 100.$

$\beta = 30.$ | $\beta = 200.$

Figure D.4: Anime samples generated with latent codes drawn from a posterior predictive density.