[Reviews · NeurIPS 2020]

Review 1

Summary and Contributions: The paper attempts to fit a generative model on the latent representations obtained from CNNs and VAEs, as a method for further studying various important properties of these. To this end, the fit a Dirichlet process Gaussian mixture model (DP-GMM), and leverage its predictive density for the aforementioned purposes. This way, the paper yields very important insights, including that networks that can exploit patterns in data learn vastly less complex representations that networks forced to memorize, that standard VAE posteriors quickly collapses to the diagonal prior when regularization strength increases, while MMD-VAEs learn more complex posterior distributions, even with strong regularization, and that fitting a DP-GMM on latent representations yields an effective sampling strategy for latent codes, improving samples quality in VAEs with rich posteriors.

Strengths: The approach is very novel and interesting. The obtained insights are important. The methodological framework is valuable and opens new avenues for further research. The empirical results are plentiful and deep.

Weaknesses: I am quite happy with the paper as it stands.

Correctness: The used DP-GMM related formulas for training and inference are pretty standard. The derivations pertaining to the used metrics are correct.

Clarity: The paper is vert well written

Relation to Prior Work: The review of prior work is satisfactory.

Reproducibility: Yes

Additional Feedback: I have read the rebuttal; it hasn't changed my view on the paper.


Review 2

Summary and Contributions: [Update after author response: I want to thank the authors for their detailed response. In my opinion the authors have convincingly addressed many questions raised by R5, and have also addressed comments 1) and 5) raised by me - my comments 2) and 3) were suggestions for further improvements but were outside the scope of this very short rebuttal phase. I continue to vote and argue for accepting the paper and stand by my score. I also want to strongly encourage the authors to add/include some of the questions raised by R5 into the discussion of the camera-ready paper - I think that many of the questions were very sensible and readers from the broader ML community might have similar questions.] The paper proposes to model neural network activation (-statistics) with a Gaussian mixture model with an unknown number of mixture components and unknown parameters for each mixture component. Both of the latter are captured via a hierarchical model (Dirichlet Process prior, and Normal-Inverse-Wishart prior) and a computationally efficient scheme for fitting the model to data and computing the posterior predictive are introduced (using a mix of closed-form solutions and sampling). The model is used to investigate the complexity of neural network representations via the KL divergence between a max-entropy reference distribution and the model posterior. In a similar fashion the model is also used to measure complexity of beta-VAE latent-space activations and quantify their disentanglement by estimating the Total Correlation. Contributions: 1) The main findings (which each are interesting contributions) of the paper are: 1a) Significant differences in representation complexity between networks that were trained on standard data compared to networks trained on random labels. 1b) Qualitative differences between data-augmentation as a regularizer compared to dropout (often dropout representations have high complexity, whereas data-augmentation more consistently has low complexity). 1c) Investigating the beta parameter in a VAE reveals that model complexity as measured by the method behaves more or less as expected (strong regularization leads to very low complexity latent representations), whereas the effect is much less pronounced for a MMD-VAE. 1d) Finally, standard VAE representations are shown to become increasingly disentangled with increasing beta, which only holds true to a weak degree for the MMD-VAE. 2) The Nonparametric model introduced in the paper is solid and sophisticated. It builds to some degree on previous work, and is well described (given space limitations) and computationally tractable ways of computing quantities of interest are provided.

Strengths: To the best of my knowledge the model proposed in the paper along with the concrete inference scheme is novel. Relations to similar previously proposed models and schemes are nicely discussed. The model is interesting and novel, and the application in the paper of investigating representational complexity is timely and interesting. Experiments are conducted in a sound fashion and results look convincing. The paper contributes towards a bigger debate regarding the nature of solutions that are obtained by training neural networks via SGD - in particular how/why solutions differ vastly in complexity when training on random/unstructured data despite no other changes in regularization strength. This debate has continuously grown over the last years and has sparked numerous pieces of interesting theoretical and empirical work, which is highly relevant to large parts of the NeurIPS community.

Weaknesses: The paper does not state current limitations of the approach. Additionally it would be nice to perform some sanity-check experiments where the ground-truth complexity or posterior predictive distribution are known (to a large degree) and the proposed method can be gauged in terms of recovering this ground-truth. The experiments are limited to image classification and image generation, which naturally limits the generalizability of the findings somewhat. And while another non-vision application would be interesting, I think that the presented experiments are already quite extensive and conducted in a sound fashion (which I personally prefer over showing more different results in a less thorough fashion).

Correctness: To the best of my knowledge the claims and methodology shown in the paper are correct. However, I cannot fully rule out that I have overlooked something in the model - particularly since complex probabilistic models with approximate inference algorithms can easily have important subtleties that are hard to spot.

Clarity: The paper is mostly well written, though another pass for typos, etc. is needed. The description of the probabilistic model and the details of the inference scheme would perhaps benefit from a slightly expanded explanation, but I personally think that the author’s already did a very good job with their explanation given the limited space of the main paper. There is no part that I would currently recommend to push to the appendix to gain more space in the main paper.

Relation to Prior Work: Related work is concisely, but nicely discussed.

Reproducibility: Yes

Additional Feedback: Overall, I greatly enjoyed reading the paper. The main questions are timely, the approach and the proposed nonparametric model are novel and interesting and so are the main findings. I have some small suggestions and comments (see below). I currently vote and argue for accepting the paper, however I will of course take into account the other reviewer’s opinions and issues raised, as well as the author’s response. I am fairly confident in my verdict, but as pointed out earlier it might be that I missed a crucial subtlety or limitation of the probabilistic model and inference scheme, which would make me reconsider my verdict. Comments: 1) Please briefly state limitations of the proposed model, as well as limitations of the main findings due to the experimental setting shown in the paper. 2) Perhaps slightly expand the discussion of choosing a maximum-likelihood fit Gaussian as the base-distribution for computing computational complexity (the KL divergence to the posterior predictive). I am happy with the maximum entropy argument, but the particular choice might explain some of the results of the VAE experiment. In particular, the second term in the VAE objective is thought to play the role of a complexity regularizer - and of course, the MMD penalty implicitly uses a different notion of complexity than the (standard) KL divergence to a isotropic Gaussian. I personally would be very convinced if the main findings would also hold qualitatively under another reasonable choice of the base-distribution for measuring complexity (though as I said before I don’t consider this as necessary for the paper, just a suggestion). 3) It would be very interesting to investigate a VAE architecture with a very flexible posterior, such as a flow-based model, and contrast the complexity of its representations against VAEs with a more constrained parametric model of the posterior. (Again, this is a suggestion, but probably beyond the scope of the rebuttal phase). 4) It would be good to discuss relations between the proposed disentanglement score and other work discussing total correlation, e.g. Auto-Encoding Total Correlation Explanation, Gao et al. (I am not one of the authors) 5) Minor comment re. Line 225 - 227: I might be missing an important subtlety, but since all conditional posteriors are regularized with an input independent prior, it should be fairly straightforward that under sufficient regularization strength the conditional posteriors should become very similar to the prior, which is an isotropic Gaussian (and thus it should not be too surprising that the complexity measure in this case goes to zero). But I don’t think there’s a problem here - I’d rather see it as an interesting sanity-check for the method.


Review 3

Summary and Contributions: The authors propose a new complexity measure for latent representations in CNNs and VAEs. They show that certain networks (CNNs with random training data, CNNs with dropout, MMD-VAEs) learn more complex representations than others (standard CNNs, standard VAEs). Update: After reading the other reviews and the author response, I have understood some things that I was missing before. I have thus increased my score.

Strengths: - The question of how complex learned representations are is interesting and has not been studied in distribution across different initializations before.

Weaknesses: - Some of the modeling choices are not perfectly motivated and not compared against other options (see below). - The main weakness is that it remains unclear whether the representational complexity is good or bad and so why exactly we should care about it in this context (see below).

Correctness: The claims and method seem to be correct. However, some of the methodological choices remain not sufficiently motivated.

Clarity: The paper is well written.

Relation to Prior Work: The prior work is properly discussed.

Reproducibility: Yes

Additional Feedback: - It is a bit unclear why the DP-GMM is necessarily the best density estimator for this task. Why not a normal GMM with a fixed number of components? - Why is the KL with the Gaussian a good complexity measure? Why not the MMD for instance? Why not use a student-t instead of a Gaussian? - In the non-parametric DP-GMM, is the number of components itself not already a measure of complexity? How does it correlate with the proposed measure? - Why is the posterior predictive a mixture of student-t distributions? - What is the relation between the proposed measure in (10) and the mutual information between dimensions? Is it the same? Why not use the total correlation instead? - Is the complexity in CNN representations good or bad? Does it correlate with performance? For instance, CNNs trained on random labels should not generalize, so I would assume the complexity is bad there. But CNNs trained with dropout usually generalize better than standard CNNs and also have a higher complexity. - Is the lack of convergence to the prior in the MMD-VAE due to the choice of kernel? Would other kernels lead to a behavior more similar to the standard VAE? - Again, is the complexity in the VAE representation good or bad? Is the MMD-VAE supposed to be the better model? From the visual quality, it's hard to tell. What are the likelihoods? - Are the results robust w.r.t. changing the random labels of the training set? How much do the representations change when the labels change?

[Author Response · NeurIPS 2020]

*Is the complexity of representations good or bad? Should we care about it? What are the limitations of your findings?*
In VAEs low-complexity aggregate posteriors are clearly bad. This is illustrated by standard $\beta$-VAEs with $\beta \geq 4$. Fig. 3 (paper) and D.3, D.4 (Appendix) show that reduced complexity translates to reduced fidelity and diversity of samples. Thus, our contribution has direct applications in development of latent variable generative models, by: 1) allowing to compare models w.r.t capacity of the latent space and independence of dimensions, 2) improving sample fidelity by fitting the aggregate posterior density. Regarding CNNs, we show that standard networks converge to a different set of solutions than memorizing nets. This adds to the evidence that neural nets exploit patterns in data and speak to the debate around Zhang et al. [ICLR 2017] work. Possible immediate applications of complexity analysis can be in interpretability research. An overview by Gilpin et al. [DSAA 2018, pp. 80–89] cites a number of works that attempt explanation by capturing semantics of network units. We uncover cases (e.g. dropout nets) where converged representations are sensitive to network initialization, making usefulness of such explanations questionable in these settings. The main limitation of our results is that we cannot claim that complexity analysis explains performance of CNNs. That said, performance-oriented engineering of deep nets vastly exceeded understanding of the proposed algorithms. Questions as basic as "are converged solutions similar?" are being answered only recently. In this context our research does contribute to the knowledge of what's happening in deep nets, even if it's not an outright explanation of generalization in deep learning.

*Why DP-GMM? Why not GMM with a fixed number of components? Main limitation of DP-GMM.* DP-GMM posterior is consistent in total variation for distributions that are in the KL support of the prior. Under smoothness assumption for the approximated density, DP-GMM yields near minimax contraction rate. See e.g. [Ghosal & van der Vaart, 2017, "Fundamentals of Nonparametric Bayesian Inference", sections 7.2 and 9.4] for details. Due to this flexibility DP-GMMs are fairly conservative choice for density estimation. A mixture with fixed number of components requires guessing the "correct" number of components. There are heuristics for this, but a more principled approach would be to use a prior on a finite number of components, i.e. a mixture of finite mixtures (MFM) model [Miller & Harrison, J. Am. Stat. Assoc, 113(2018), pp. 340-356]. While there could be some merit in doing so, these models are more restrictive than DP-GMMs (see below). The main limitation of Gaussian mixtures is computational cost in very high-dimensional spaces – this must be taken into account when constructing neural representations.

*Number of components as a measure of complexity.* Number of components can be seen as a measure of complexity – sample complexity of learning a Gaussian mixture is linear in the number of components [Ashtiani et al., NeurIPS 2018, pp. 3416–3425]. But there are caveats. DP is a prior on infinite mixtures and will not concentrate on a finite number of components in the infinite data limit [Miller & Harrison, JMLR, 15(2014), pp. 3333-3370]. We can get consistency for the number of components with an MFM model. A more fundamental issue is that finite mixture models (including GMM with a fixed number of components) make sense only if the true data generating distribution is a finite mixture. To reason about the number of components we also need to know component distributions. Unless one already has a good understanding of the data generating process, these are fairly strong assumptions. The appeal of density analysis via infinite (in the limit) mixtures is that we can avoid making such assumptions.

*Why is the KL with the Gaussian a good complexity measure? Why not MMD or Student's t-distribution?* Basically, we choose reference distribution following maximum entropy principle. We pick distribution that encodes mean and variance of the data, but otherwise minimizes additional assumptions. Under maximum entropy principle this will be a Gaussian. It may seem that a reasonable alternative could be a maximum entropy distribution that exactly fits the support of the data (uniform distribution). However, the reference distribution and the posterior predictive would then have different supports, leading to problems with divergences (its unreasonable to restrict the posterior to the support of known data). T-distribution has less obvious justification – we could use it to measure divergence between prior and posterior predictive in DP-GMM. Reviewer #3 points out that the base distribution may coincide with the prior in VAEs with strong regularization. We actually leverage this to claim that under strong regularization posteriors in $\beta$-VAEs collapse to the prior (we will make this more explicit in the text). These results can also be seen as a sanity check for our model (see also third paragraph in Appendix D). KL divergence has intuitive interpretations in information theory. Further, it is much more common to reason about e.g. total correlation than distances between kernel mean embeddings.

*Why is posterior predictive a mixture of t-distributions? Eq. 10 and total correlation.* To be precise, it's the posterior predictive given component assignments that is a mixture of t-distributions – detailed derivation is in Appendix B. Eq. 10 is an estimate of the total correlation between dimensions in the posterior predictive – we will make that explicit in the text.

*Lack of convergence to the prior in MMD-VAE. Is the MMD-VAE supposed to be the better model?* In addition to IMQ kernel (used in the original MMD-VAE paper) we also experimented with an RBF kernel and did not observe substantial differences in results. Evaluating other kernels is quite interesting, but may fall outside the scope of this paper. Zhao et al. [AAAI-19, pp. 5885–5892] reports higher likelihoods for MMD-VAE.

*Robustness of results w.r.t. changing the random labels.* We use two datasets in CNN experiments. They have different labels and were permuted independently. Results for both datasets are compatible. Also, in initial experiments we did not fix random seeds and did not observe different outcomes due to specific label permutations.

[Meta-Review · NeurIPS 2020]

The paper models neural network activation statistics with a Gaussian mixture model with an unknown number of components. The model is used to investigate the complexity of representations through the KL divergence between the max entropy reference and the model posterior. The reviewers generally felt the paper made a variety of interesting observations. Congratulations on the nice work. In a final version, the authors are encouraged to read and account for updates to reviewer comments after the rebuttal, and to discuss https://arxiv.org/abs/2002.08791, which provides a complementary Bayesian nonparametric perspective on deep neural networks.